



# Climate and geology overwrite land use effects on soil organic nitrogen cycling on a continental scale

Lisa Noll[1,2], Shasha Zhang[1], Qing Zheng[1], Yuntao Hu[1,3], Florian Hofhansl[4] and Wolfgang Wanek[1*]

[1]Division of Terrestrial Ecosystem Research, Department of Microbiology and Ecosystem Science, Center of Microbiology
and Environmental Systems Science, University of Vienna, Vienna, Austria
[2]German Environment Agency, Dessau-Rosslau, Germany
[3]Lawrence Berkeley National Laboratory, Berkeley, USA
[4]International Institute for Applied Systems Analysis, Schlossplatz 1, A-2361 Laxenburg, Austria

*Correspondence to*: Wolfgang Wanek (wolfgang.wanek@univie.ac.at)

**Abstract.** Soil fertility and plant productivity are globally constrained by N availability. Proteins are the largest N reservoir in soils and the cleavage of proteins into small peptides and amino acids has been shown to be the rate limiting step in the terrestrial N cycle. However, we are still lacking a profound understanding of the environmental controls of this process. Here we show that integrated effects of climate and soil geochemistry drive protein cleavage across large scales. We measured gross protein depolymerization rates in mineral and organic soils sampled across a 4000-km-long European
transect covering a wide range of climates, geologies and land uses. Based on structural equation models we identified that soil organic N cycling was strongly controlled by substrate availability e.g. by soil protein content. Soil geochemistry was a secondary predictor by controlling protein stabilization mechanisms and protein availability. Precipitation was identified as the main climatic control on protein depolymerization by affecting soil weathering and soil organic matter accumulation. In contrast, land use was a poor predictor of protein depolymerization. Our results highlight the need to consider geology and
precipitation effects on soil geochemistry when estimating and predicting soil N cycling at large scales.

## 1 Introduction

Microbial decomposition of soil organic matter is a fundamental driver of soil ecosystem functions and services e.g. nutrient regeneration through decomposition maintains soil fertility and plant productivity. For example, the extracellular cleavage of plant- and microbial-derived soil proteins, chitin or peptidoglycan to small organic compounds such as peptides, amino acids
and amino sugars regulates organic N uptake by soil microbes, contributes to plant N nutrition and further drives terrestrial inorganic N cycling (Hu et al., 2018; Noll et al., 2019b). Proteins account for up to 90 % of soil N (Martens and Loeffelmann, 2003; Schulten and Schnitzer, 1997). Protein depolymerization is mediated by extracellular enzymes and facilitates microbes and plants to utilize the by far single largest N reservoir in soils. However, the large-scale controls of gross protein depolymerization are largely unknown. Since protein depolymerization is mediated by extracellular enzymes it
is tied to substrate availability, soil geochemistry and microbial N demands driving enzyme production (Sinsabaugh et al.,



2008). Across biogeographic regions peptidase activity increases strongly with soil pH since the pH optima of most proteolytic enzymes is about 7 – 8 (Sinsabaugh et al., 2008; Hendriksen et al., 2016). Moreover, soil pH is a major control on bacterial community composition and cross-continental studies showed that this pattern is consistent across soil types and biomes (Lauber et al., 2009; Rousk et al., 2010; Fierer and Jackson, 2006). Given the large difference in the excreted enzyme complement between microbial taxa, soil nutrient status and edaphic properties (e.g. soil pH, texture and cation exchange capacity) shape the set of excreted proteolytic enzymes (Lauber et al., 2009; Lauber et al., 2008; Jangid et al., 2008; Fuka et al., 2008) by their effects on microbial community composition. Effects of climate on peptidase activity are rather indirect, indicated by shifts in vegetation type and in soil nutrient stoichiometry from low to high latitudes (Hendriksen et al., 2016; Sinsabaugh et al., 2008; Peng and Wang, 2016). Soil C:N ratios  are typically higher in forest soils than in agricultural soils, and affect in particular the fungi: bacteria ratios (Lauber et al., 2008). Land use can consequently affect the production of soil extracellular enzymes (Jangid et al., 2008).

Substrate availability is likely the most striking control on organic N depolymerization rates and has been shown to be driven by land use and soil properties at the regional scale (Noll et al., 2019b). Soil N stocks as proxy for protein contents are typically increasing with mean annual precipitation and to decrease with the level of aridity (Delgado-Baquerizo et al., 2013; Marty et al., 2017; Callesen et al., 2007). Changes in temperature and precipitation patterns are associated with changes of the potential natural vegetation, where N becomes progressively limiting with vegetation changes from deciduous to coniferous shrubs and trees and from low to high latitudes (Kang et al., 2010; Reich and Oleksyn, 2004). Moreover, soil N stocks decrease with intensification of land management, from forests to grasslands and croplands (Six and Jastrow, 2002). Decomposition experiments of plant litter and organic soils showed an inverse relationship of gross protein depolymerization rates and resource C:N ratios and a positive with N content, suggesting that protein depolymerization is rather controlled by substrate availability than by the pool size of extracellular enzymes (Mooshammer et al., 2012). However, in mineral soils this relationship was less pronounced, indicating that protein stabilization on mineral surfaces may restrict soil protein cleavage (Wild et al., 2013; Noll et al., 2019b).

In mineral soils, organic nitrogen availability is constrained by interactions of organic compounds with the soil matrix e.g. formation of organo-mineral associations, and restricted accessibility in small pores and soil aggregates render soil organic matter protected from enzymatic attack (Kögel-Knabner et al., 2008; Quiquampoix, 2000). Stabilization mechanisms are controlled by soil texture and soil mineral assemblage, and particularly by the amounts of Fe- and Al oxyhydroxide which are major sorption sites of soil organic matter in soils (Kaiser and Guggenberger, 2000). Their amount and composition are shaped by soil parent material (primary minerals) and environmental conditions during pedogenesis controlling bedrock weathering and the formation of secondary minerals. Both, protein availability and proteolytic activity are further constrained by the formation of metal-organic complexes or complexation with tannins (Nierop et al., 2002; Peter J Hernes, 2001; Adamczyk et al., 2009).

Land use, bedrock and biogeographic region are therefore key controls on soil nutrient status and edaphic properties and affect microbial community structure, substrate availability and microbial N and C demands (Figure 1). Changes in




environmental conditions might thereby be translated into altered organic N process rates. To investigate the major controls on organic N cycling, we sampled a large-scale transect across Europe from the Mediterranean to the Subarctic, covering three different land use types (forest/shrub land, grassland and cropland) as well as a wide range of climates and geologies and determined gross protein depolymerization rates using an isotope pool dilution approach targeting soil amino acid production (protein depolymerization).

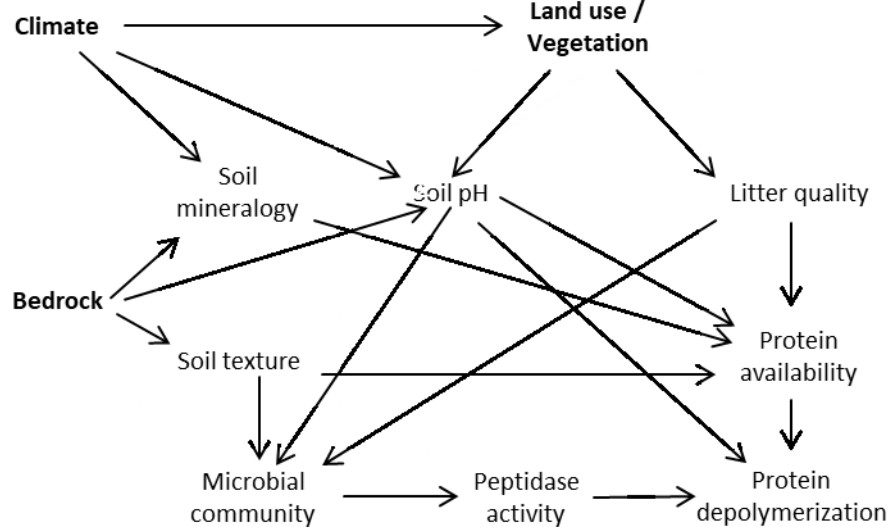

**Figure 1 Proposed model relating climate, bedrock and land use effects to protein depolymerization rates.**


We hypothesized that (I) protein depolymerization is restricted by the lower soil organic matter content and microbial activity in cropland soils compared to grassland and forest soils. (II) We further expected that the availability of proteins and thereby gross protein depolymerization rates are controlled by soil geochemical properties (e.g. soil pH), mineral assemblage

and texture. (IV) We further hypothesized that climate is a rather indirect control on organic N cycling by its effects on vegetation and soil geochemistry as well as on soil N stocks.

## 2 Materials and Methods

### 2.1 Sampling

Soil samples were collected during summer 2017 (May to August) at the peak of the growing season across a European

continental transect from the warm Mediterranean to the cold Subarctic and from the humid Atlantic western climate to the dry continental steppes in Romania (Figure 2). The sampled soils were distinct in soil parent material, soil type, land use and vegetation. Sampling sites were selected to represent the natural vegetation as defined in the 'Map of the natural vegetation of Europe' (Bohn and Katenina, 2000). For each sampling site climate data scaled to 100 m were extracted from the WorldClim database vs. 1.4 (Fick and Hijmans, 2017). Bedrock was obtained from the international geological map of



Europe (IGME5000, 1:5.000.000 *(Asch, 2005)*) and dominant soil types were obtained from the "Soil regions of European Union and adjacent areas" map (EUSR5000, 1:5.000.000, (Bgr [Bundesanstalt Für Geowissenschaften Und Rohstoffe], 2005).

For statistical analyses bedrock types were binned into three groups: limestone, sediment and silicate. Sediment geologies included Flysch, Molasse, till and fluvial sand, silicate bedrock included plutonic, igneous and metamorphic formations, and

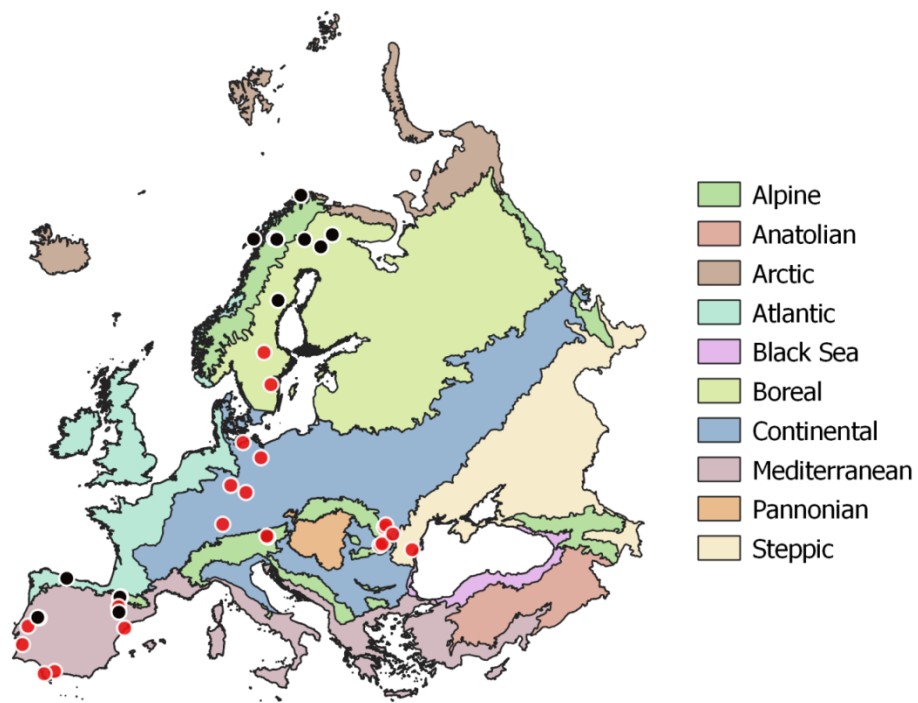

**Figure 2 Sampling sites across European biogeographical regions. Red circles symbolize sampling sites including three land use types (woodland, grassland, cropland). Map of European biogeographical regions was obtained from biogeographical regions dataset of the European Environment Agency.**

carbonate bedrock ranged from dolomite to limestone and marl. Mean annual temperature (MAT) of the sampling sites ranged from -3.5 to 17.8 °C and mean annual precipitation ranged from 415 to 1396 mm y$^{-1}$. Where possible, all three management types (woodland/forest, grassland and cropland) as well as mineral and organic soils were sampled in close vicinity. In the following we only use 'woodland' for subarctic tundra, open woodlands and forests. At each site bulk samples of mineral top soil (0-15 cm) were taken with a soil corer (5 cm). Each bulk soil sample consisted of five replicates

with about 5 m distance from each other. In total we sampled 96 mineral top soils from 43 sites. 23 sites included woodland, grassland and cropland soils (Table S1). Organic layers were sampled at 13 sites using a 20 x 20 cm frame to cut out the organic horizon down to the mineral soil surface. Representative leaf litter samples were collected at each site and represent





the dominating vegetation. Roots and stones were removed from the soil samples manually immediately after sampling. Soil samples, roots, stones and litter samples were cooled (4-8 °C) and shipped within 3 to 7 days to the University of Vienna for

further analyses. Soil samples were homogenized by sieving to 2 mm and separate aliquots were air dried or stored moist at 4 °C. Litter and root samples were washed and dried in a drying oven at 60 °C.

## 2.2 Basic soil parameters

Soil texture, $CaCO_3$ content, cation exchange capacity (CEC), base saturation (BS) and exchangeable $Ca^{2+}$, $Mg^{2+}$, $K^+$, $Na^+$, $Al^{3+}$, $Fe^{3+}$ and $H^+$ were determined by the Austrian Agency for Health and Food Safety (AGES) according to European and

international standards (ÖNORM). Fe- and Al-oxyhydroxides were determined in acid ammonium oxalate and in Na-dithionite extracts (Loeppert, 1996) at the Institute of Soil Research (IBF, University of Natural Resources and Life Sciences, Vienna, Austria). Oxalate extractable Fe ($Fe_{oxalate}$) and Al ($Al_{oxalate}$) refers to amorphous Fe- and Al oxyhydroxides and Fe bound in organo-metal complexes. Dithionite extractable Fe minus oxalate extractable Fe represents Fe bound in crystalline oxyhydroxides ($Fe_{d-o}$). The ratio of oxalate extractable Fe over dithionite extractable Fe presents a measure of the

activity of the Fe-mineral phase ($Fe_{o/d}$). To determine the soil water content, sieved soils were dried at 85 °C for 48 h. Water holding capacity (WHC) was measured by repeatedly saturating 10 g field-moist soil with deionized water and draining in between for 2.5 hours in a funnel with an ash-free cellulose filter paper. Field-moist soils were either adjusted to 60% WHC by gentle drying at room temperature or by addition of deionized water. Before further analyses all soils were pre-incubated for two weeks at 20 °C and 60% water holding capacity (WHC) in PE-Ziploc bags. Soil pH was measured in water and 10

mM $CaCl_2$ (1 : 5 (w : v)) using an ISFET pH sensor (Sentron, Leek, The Netherlands). To determine total C and total N in root and litter as well as soil organic C (SOC) and soil total N (TN) oven dried root, litter and soil samples were ground with a ball mill (MM 200, Retsch, Germany) and analyzed by an Elemental analyzer (Carlo Erba 1110, CE Instruments) coupled to a Delta$^{Plus}$ Isotope Ratio Mass Spectrometer (Finnigan MAT, Germany) via a Conflo III interface (Thermo Fisher, Austria). If necessary, carbonates were removed from soil samples with 2 M HCl prior to SOC and TN measurements. Soil

total P (TP) and soil total inorganic P (TIP) were determined in 0.5 M $H_2SO_4$ extracts of ignited (450 °C, 4 °C (Lajtha et al., 1999)) and control soil aliquots followed by malachite green measurements of reactive phosphate (Kuo, 1996). Total soil organic P (TOP) was calculated as the difference of TP – TIP. Soils were extracted with 1 M KCl (1:5 (w:v)) for 1 h and filtered through ash-free cellulose filters (Whatmann). Dissolved organic C (DOC) and total N (TDN) were measured in the extracts by a TOC/TN analyzer (TOC-VCPH/TNM-1, Shimadzu, Austria). $NH_4^+$ and $NO_3^-$ were measured colorimetrically

in the same extracts (Hood-Nowotny et al., 2010). Dissolved organic N (DON) was calculated as TDN minus $NO_3^-$ and $NH_4^+$. Free amino acids (FAA) were determined fluorimetrically in 1 M KCl extracts by the OPAME fluorescence method (Jones et al., 2002) as modified by Prommer et al. (2014). Dissolved inorganic P (DIP, Olsen P) was extracted with 0.5 M $NaHCO_3$ (1 : 7.5 (w : v), pH 8.5) for 1 h, filtered through ash free cellulose filters and measured by malachite green. Total dissolved P (TDP) was measured following acid persulfate digestion and dissolved organic P (DOP) was calculated as the

difference of P concentration between digested and non-digested samples (Lajtha et al., 1999). Soil microbial community





composition was analyzed by phospholipid fatty acid (PLFA) analyses according to Kaiser et al. (2010) and Hu et al. (2018). Microbial C, N and P were determined by chloroform fumigation extraction (Brookes et al., 1985). Sample aliquots were fumigated for 48 h and subsequently extracted as described above with 1 M KCl or 0.5 M NaHCO$_3$. Potential activities of leucine-amino peptidase (EC 3.4.11.1) was determined in buffered (Na-acetate, pH 5.5) and unbuffered (ultra-pure water)
soil slurries using L-leucine-7-amido-4-methyl coumarin (AMC-leucine) as substrate (Kaiser et al., 2010). Triplicates of each sample were incubated for 2 h at 25 °C and measured every 30 min. Fluorescence was measured with a TECAN InfiniteR M200 (Austria) spectrophotometer at an excitation wavelength of 365 nm and an emission wavelength of 450 nm, and was corrected for sample blank fluorescence and quenching prior to calculations of AMC concentration.

## 2.3 NaOH extractable protein

2 g of fresh soil were extracted with 0.5 M NaOH (1 : 10 (w : v)) for 2 h in an ultra-sonic bath (160 W, Sonorex RK510, Germany) and subsequently for further 16 h on a rotary shaker. NaOH extracts free and loosely bound proteins e.g from organo-mineral associations but not proteins stabilized in metal-organo complexes (Wattel-Koekkoek et al., 2001). Extracts were centrifuged for 15 min at 1600 x g. As high salt concentrations interfere with the consequent measurement of hydrolyzed amino acids, 2.5 ml of supernatant were desalted using Sephadex™ G-25 columns (PD10 GE Healthcare,
Uppsala, Sweden). For determination of total amino acids we adopted a method published by Martens and Loeffelmann (2003) and Hu et al. (2018). The purified extracts were freeze-dried and re-dissolved in 1.5 ml methanesulfonic acid (4 M MSA). 1 ml of samples, bovine serum albumin (BSA) standards, and blanks were hydrolyzed in an autoclave for 1 h at 135 °C. Hydrolyzed extracts were neutralized with 4 M KOH and measurements were performed on an HPLC system (Dionex ICS-3000, Thermo Fisher Scientific, Bremen, Germany) coupled to an electrochemical detector. Amino acids were separated
using a PA-10 IC column (Thermo Fisher Scientific, Bremen, Germany). NaOH-extractable protein (Protein$_{NaOH}$) was calculated as the sum of the 20 measured amino acids.

## 2.4 Gross organic N processes

One day before starting the pool dilution experiment FAA concentrations were determined in an aliquot of pre-incubated soil. The isotope pool dilution experiment and sample analyses were conducted as described previously by Noll et al.
(2019a). In brief, 4 g of soil were weighed into transparent HDPE vials in duplicates and 400 μl of a $^{15}$N tracer solution were added drop wise. Samples were shaken vigorously to guarantee good mixing of the tracer. The tracer solution was prepared from a highly $^{15}$N enriched amino acid mixture (U-15N-98 at% $^{15}$N amino acid mixture from crude algal protein, Cambridge Isotope Laboratories, Radeberg, Germany). The total amount of added $^{15}$N was adjusted to about 20% of the native FAA pool. The incubation was terminated after 15 and 45 min by addition of cold KCl (4 °C) and samples were extracted for 1 h
on a rotary shaker and filtered at 4 °C. Prior to measuring the isotopic composition of FAA NH$_4^+$ was removed by microdiffusion Lachouani et al. (2010); (Noll et al., 2019a). Extracts were microdiffused for 48 h. To measure the concentration and atom %$^{15}$N of FAA 2 ml of pre-treated extracts were transferred into 12 mL glass exetainers and the α-





amino group was cleaved/oxidized by NaClO and KBr as catalyst under alkaline conditions as described by Zhang and Altabet (2008) and modified by Noll et al. (2019a). Subsequently the produced $NO_2^-$ was converted to $N_2O$ by buffered

$NaN_3$ ($NaN_3$ in 100% acetic acid 1:1). The produced $N_2O$ was measured with a purge-and-trap isotope ratio mass spectrometer (PT-IRMS) consisting of a Finnigan Delta V Advantage IRMS (Thermo Fisher, Germany) and a Gasbench II headspace analyzer (Thermo Fisher, Germany) with cryo-focusing unit. Calibration was done according to Lachouani at al. (2010) and Noll et al. (2019a)

**2.5 Data analyses and statistics**

Gross rates of protein depolymerization (GP) and microbial amino acid uptake (GU) were calculated according to Kirkham and Bartolomew (1954) and Wanek et al. (2010):

$$GP = \frac{(N_{t2} - N_{t1})}{(t2 - t1)} * \frac{LN\left[\frac{(at\%^{15}N_{t1} - at\%^{15}N_b)}{(at\%^{15}N_{t2} - at\%^{15}N_b)}\right]}{LN\left(\frac{N_{t2}}{N_{t1}}\right)}$$

$$GU = \frac{(N_{t1} - N_{t2})}{(t2 - t1)} * \left(1 + \frac{LN\left[\frac{(at\%^{15}N_{t2} - at\%^{15}N_b)}{(at\%^{15}N_{t1} - at\%^{15}N_b)}\right]}{LN\left(\frac{N_{t2}}{N_{t1}}\right)}\right)$$

where $N_{t1}$ and $N_{t2}$ are the concentrations of FAA-N at the time points t1 (15 min) and t2 (45min). $^{15}N$ content in amino acids at the time points of termination are expressed as $at\%^{15}N_{t1}$ and $at\%^{15}N_{t2}$, while $at\%^{15}N_b$ is the background $^{15}N$ abundance

(0.366 $at\%^{15}N$) in non-labeled samples. Mean residence times of FAA were estimated as free amino acid pool size divided by microbial amino acid uptake rate. Microbial C:N and N:P imbalances were calculated as the ratio of resource C:N or N:P over microbial C:N or N:P.

For statistical analyses mineral soils were grouped by bedrock (limestone, sediments, silicates) or by land use (cropland, grassland, woodlands). Prior to statistical analyses data were checked for normality and transformed if necessary. Effects of

bedrock were analyzed by one-way analysis of variance (ANOVA) followed by Tukey HSD tests. To analyze land use effects on process rates and soil properties site was included as factor in a two-way ANOVA to account for differences between sites (climate, bedrock, soil type). Land use effects were only analyzed for 22 sites where cropland, grassland and woodland soils could be sampled in close vicinity. Differences in process rates and soil properties between organic and underlying mineral soil horizons were analyzed by paired t-tests for the 13 sites where organic and mineral horizons were

sampled. Linear mixed models were used to explore the effect of soil properties and climate on protein depolymerization rates with land use as random factor. The most parsimonious model was selected by Akaike's information criterion (AIC). Multicollinearity was assessed by variance inflation factors (VIF). Variables with VIF larger than 2.5 were excluded from the model. Partial correlations were used to control for the effect of soil geochemical properties on the relationship between





climate and the response variables (i.e. protein depolymerization rates, leucine-amino peptidase activity and NaOH-
extractable Protein; (Doetterl et al., 2015; Luo et al., 201). Significant changes of the correlation coefficient were assumed
when the 95% confidence interval of the zero-order correlation and the partial correlation did not overlap. Partial correlations
were analyzed using 'ppcor' in R environment (Kim, 2015). Effects of climate parameters and their interactions on process
rates were assessed by linear mixed effect models with soil parent material or land use as random effects. We used structural
equation modelling (SEM) to explore direct and indirect effects of climate, geology and soil properties on protein
depolymerization rates. We used parameters which correlated significantly with protein depolymerization to construct a base
model for gross protein depolymerization rates. Input variables were tested for multivariate normality and linearity. If
necessary variables were log transformed to mitigate departure from model assumptions. The model was then analyzed using
the 'lavaan' package (Rosseel, 2018) in R. Model fit was evaluated using Chi-square statistics ($p > 0.05$). The most
parsimonious model was identified by step-wise deletion of non-significant paths. Akaike's information criterion (AIC) was
used to compare competing model fits. We followed the two-index strategy proposed by Hu and Bentler (1999) to describe
the specified model and the data covariance-matrix and reported root mean square error of approximation (RMSEA) and
standardized root mean square residual (SRMR). Good model-data fit is indicated by RMSEA $\leq 0.06$ and SRMR $\leq$
0.08. All statistics were performed in R 3.1.3 (R Development Core Team, 2008).

## 3 Results

### 3.1 Effects of bedrock, land use, soil horizon and climate

Soil properties, leucine-amino peptidase activity, gross protein depolymerization, FAA uptake and MRT of FAA were
analyzed for differences between mineral soils of the three major bedrock classes (limestones, sediments and silicates) and
land use types (cropland, grassland, woodland), as well as for differences between the two soil horizons (mineral, organic,
Table S2). Bedrock had significant effects on soil physicochemical properties but not on root biomass, root C:N and $NO_3^-$.
Soil pH measured in water was highest in limestone soils ($7.9 \pm 1.1$) followed by sediment soils ($6.0 \pm 1.1$) and silicate soils
($5.3 \pm 0.9$). Clay content increased in the order silicate<sediment<limestone. Limestone soils were characterized by higher
$Fe_{d-o}$, representing crystalline Fe oxides, ($13.6 \pm 10.5$ mg g$^{-1}$) but lower $Fe_{oxalate}$ ($4.3 \pm 0.1$ mg g$^{-1}$) and $Al_{oxalate}$ ($2.9 \pm 1.6$ mg g$^{-1}$)
contents, representing amorphous Fe- and Al- (hydro)oxides, than sediment and silicate soils and geochemical parameters
were always significantly different from silicate soils (Table S2). Soil C (SOC), total N (TN) and P (TP) contents increased
in the order sediment<limestone<silicate soils and were lowest in sediment soils (SOC: $22.6 \pm 15.3$ mg g$^{-1}$, TN: $3.4 \pm 2.4$ mg g$^{-1}$, TP: $0.5 \pm 0.3$ mg g$^{-1}$) than in silicate soils (SOC: $43.0 \pm 31.5$ mg g$^{-1}$, TN: $2.6 \pm 1.6$ mg g$^{-1}$, TP: $0.7 \pm 0.4$ mg g$^{-1}$). Soil C:N was
higher in silicate soils ($18.1 \pm 7.3$) than in sediment ($11.8 \pm 3.0$) and limestone ($11.6 \pm 2.9$) soils. Microbial C was also strongly
affected by bedrock, with lower $C_{mic}$ in sediment soils ($0.6 \pm 0.3$ mg g$^{-1}$) than in limestone ($1.0 \pm 0.6$ mg g$^{-1}$) and silicate
($1.1 \pm 0.8$ mg g$^{-1}$) soils. Bedrock had significant effects on bacterial PLFA contents but not on fungal PLFA content and the





ratio of bacterial to fungal PLFA. Microbial C:N imbalance was lower in sediment soils than in limestone and silicate soils and microbial N:P imbalance was lower in sediment and silicate soils than in limestone soils. NaOH-extractable protein and FAA increased in the order limestone = sediment < silicate. In limestone soils NaOH-extractable protein was lower ($68.5\pm55.5$ µg N g$^{-1}$) than in sediment ($80.8\pm47.6$ 5 µg N g$^{-1}$) and silicate ($146.9\pm107.6$ 5 µg N g$^{-1}$) soils. This corresponds to lower protein depolymerization rates in limestone ($32.2\pm23.9$ µg N g$^{-1}$ d$^{-1}$) and sediment soils ($51.3\pm58.7$ µg N g$^{-1}$ d$^{-1}$) than in silicate soils ($181.5\pm172.6$ µg N g$^{-1}$ d$^{-1}$) (Figure S2). Highly similar patterns to those of protein depolymerization were observed for FAA-N uptake (Figure S1). In contrast, leucine-amino peptidase activity was significantly higher in limestone soils ($0.93\pm0.74$) than in sediment ($0.29\pm0.49$ µmol g$^{-1}$ h$^{-1}$) and silicate ($0.18\pm0.27$ µmol g$^{-1}$ h$^{-1}$) soils. Mean residence times (MRT) of FAA were not varying significantly between the three bedrock types (Figure S2).

Effects of land use on the organic N cycle were analyzed for 22 sites comprising woodland, grassland and cropland mineral soils (Table S2). Soil pH was significantly lower in woodland soils ($6.0\pm1.5$) than in grassland ($6.7\pm1.3$) and cropland soils ($6.9\pm1.2$). Root biomass and root C:N ratio as well as SOC, DOC, microbial C and bacterial biomass (PLFA$_{gram+}$, PLFA$_{gram-}$, PLFA$_{Bacteria}$) were significantly affected by land use and lower in cropland soils compared to woodland soils. CEC, Al$_{oxalate}$, Fe$_{oxalate}$, Fed-o, fungal biomass (PLFA$_{fungi}$) and fungi:bacteria PLFA ratio (PLFA$_{fungi:bacteria}$) were not affected by land use. We found no significant effect of land use on soil total N content, but proportions of dissolved organic and inorganic N species varied with land use (Figure S4). DON accounted for 61% of total dissolved N in grassland soils and 50% in woodland soils. In contrast, in cropland soils dissolved N was dominated by NO$_3^-$ (69%). FAA-N was significantly lower in cropland soils ($1.4\pm0.64$ µg N g$^{-1}$) than in grassland ($4.15\pm5.16$ µg N g$^{-1}$) and woodland soils ($3.23\pm3.71$ µg N g$^{-1}$) (Table S2). The proportion of FAA-N in DON increased in the order cropland (21.9%) < grassland (32.0%) = woodland (33.0%). NaOH-extractable protein showed significant differences between cropland and woodland soils and decreased in the order woodland soils ($101.7\pm47.4$ µg N g$^{-1}$) > grassland ($93.8\pm41.3$ µg N g$^{-1}$) > cropland ($58.6\pm52.3$ µg N g$^{-1}$) soils. Gross protein depolymerization rates were significantly lower in cropland soils ($32.9\pm27.8$ µg N g$^{-1}$ d$^{-1}$) compared to grassland ($83.7\pm124.4$ µg N g$^{-1}$ d$^{-1}$) and woodland soils ($72.2\pm79.6$ µg N g$^{-1}$ d$^{-1}$) (Figure S1). Leucine-amino peptidase activity increased in the order cropland=grassland>woodland, however only woodlands ($0.34\pm0.52$ µmol g$^{-1}$ h$^{-1}$) were significantly lower than grasslands ($0.61\pm0.78$) and croplands ($0.46\pm0.6$ µmol g$^{-1}$ h$^{-1}$) (Figure S1, Table S2). Mean residence times of FAA were not affected by land use type (Figure S3, Table S2).

At 13 forest sites we sampled the corresponding organic horizons. Differences between the organic horizons and the underlying mineral soil were analyzed by paired t-tests (Table S2). Soil pH was significantly lower in organic soil horizons than in mineral soil horizons. Soil bacterial and fungal PLFA contents were not significantly different between mineral and organic soil horizons. CEC as well as soil C (SOC, DOC), N (total N, DON, NH4+, NO3-, FAA-N) and P (TP, DOP, TOP) contents were higher in organic soil horizons than in mineral horizons. Organic horizons showed significantly higher NaOH-extractable protein contents ($347.6\pm58.9$ µg N g$^{-1}$) than mineral soil layers ($159.1\pm122.4$ µg N g$^{-1}$). Leucine-amino peptidase activity, protein depolymerization and FAA-uptake were significantly higher in organic soil layers than in the underlying mineral soil layers. Protein depolymerization rates in organic layers ($926.9\pm656.0$ µg N g$^{-1}$ d$^{-1}$) exceeded those in mineral





layers (167.45±159.5 µg N g$^{-1}$ d$^{-1}$) by 5-fold. Leucine-amino peptidase activity was 6-fold higher in organic soil layers

(0.29±0.11 µmol g$^{-1}$ h$^{-1}$) than in mineral soil layers (0.05±0.02 µmol g$^{-1}$ h$^{-1}$).

Correlation analyses of protein depolymerization showed positive relationships with amorphous Fe and Al minerals (Fe$_{oxalate}$, Al$_{oxalate}$), soil organic matter (Corg, total N), NaOH-extractable protein and microbial biomass (Cmic, PLFA) (Figure 3, Table S3). NaOH extractable protein content increased with SOC, TN, root biomass and amorphous Fe- and Al - (hydr)oxides (Table S3, Figure 3). Soil pH was negatively correlated with gross depolymerization and NaOH-extractable

protein but positively to peptidase activity (Figure 3). Peptidase activity was positively related to microbial C, CEC, clay and silt content. However, across all sites as well as within subgroups we found no significant correlation between peptidase activity and protein depolymerization rates (Figure S5). In order to further examine the potential edaphic controls on protein depolymerization rates in mineral soils as well as interaction effects with land use we used multiple linear regression

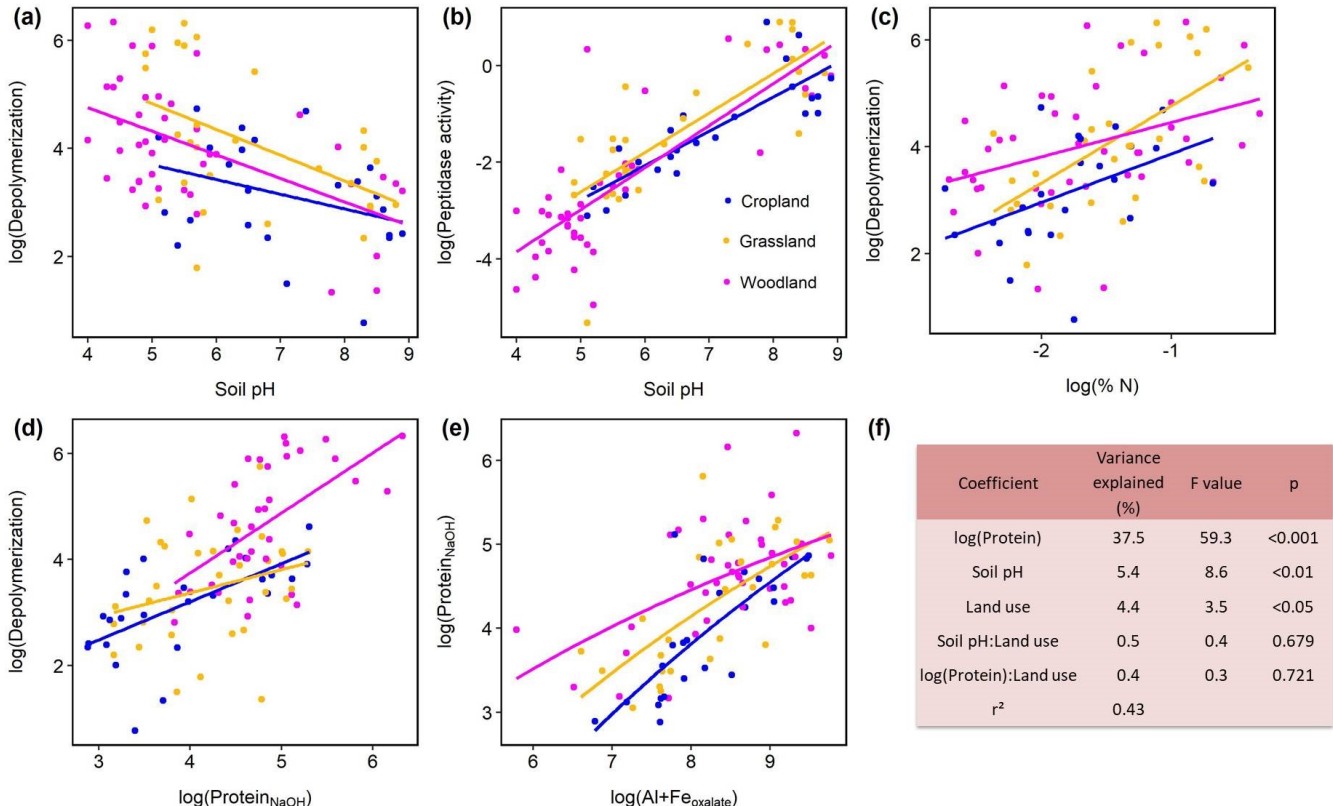

**Figure 3 Effects of soil properties on gross protein depolymerization rates in mineral soils. Relationship of pH and (a), log(Protein depolymerization) and (b), log(leucine-amino peptidase activity). (c), Relationship of soil total N and protein depolymerization rate (d), Relationship of NaOH-extractable protein and protein depolymerization rate. Color codes indicate land use type. (e), Relationship of oxalate extractable Al and Fe and NaOH-extractable protein. (f), Analyses of variance of the most parsimonious linear regression model of log(gross protein depolymerization rate) explained by soil properties, land use and their interaction effects (n = 95). Total model fit is given as adjusted r².**



analyses. The most parsimonious model included a strong positive effect of NaOH-extractable protein explaining 37% of the
variance (Figure 3). Soil pH had a negative effect on depolymerization rates and parameter estimates for land use increased
in the order cropland<grassland<woodland. Interaction effects between edaphic properties and land use were not significant.
Linear mixed effect models with land use as random factor revealed the same main predictors and the final model explained
69% of the variation (Table S4).

Climate effects on depolymerization rates were analysed by linear regression analyses including climate parameters,
land use and interaction effects. We found significant effects of MAT, MAP and their interaction effect (MAP:MAT) (Table
S5). Land use had no significant effect, as shown by the strong negative correlation between depolymerization and MAT
across the three studied land use types (Figure 4). The model explained about 42% of the variance. Although, the climatic
humidity index (MAP:PET) expressed as MAP over potential evapotranspiration (PET) was not included in the most
parsimonious model the strong non-linear increase of depolymerization rates with climatic humidity ($r^2$=0.632, p<0.001)
across all sites and land use types was striking (Figure 4). The most parsimonious linear mixed effect model included land
use as random factor and showed a strong negative effect of MAT as well as a positive effect of MAP. The model explained
about 47% of the variance. The interaction of MAT:MAP was not significant.

## 3.2 Interactive effects of edaphic properties and climate

Since soil parent material, which is a main driver of soil geochemical properties, is not uniformly distributed across the
sampled transect, climate effects (MAT and MAP) on gross protein depolymerization rates, leucine-amino peptidase activity
and NaOH-extractable protein were analysed by partial regression analyses controlled for geochemical parameters (Figure
4). We found a negative zero-order correlation between protein depolymerization rates and MAT (r = -0.63, p<0.01). A
significant decrease of the correlation coefficient was observed when removing correlations with Al, Fe or the sum of
oxalate extractable Fe and Al (Figure 4). Leucine-amino peptidase was positively correlated to MAT (r = 0.34, p<0.05).
After removing the correlations with soil pH or Al we found a negative correlation between peptidase activity and MAT.
Removal of the correlation with soil P content significantly increased the positive correlation between peptidase activity and
MAT. NaOH-extractable protein was negatively correlated to MAT (r = -0.53, p<0.01). The correlation coefficient was
significantly decreased by removing the correlations with Al and the sum of oxalate extractable Al and Fe. All zero-order
correlations with MAT decreased significantly after removing the effects of all used soil geochemical parameters. Mean
annual precipitation was weakly positively correlated with protein depolymerization rates (r = 0.29, p<0.05). The removal of
correlations with geochemical parameters did not change the correlation coefficient significantly. Leucine-amino peptidase
activity was not correlated to MAP; however the removal of soil pH or Al increased the correlation coefficient significantly,
inducing a weak positive correlation between leucine-amino peptidase activity and MAP. NaOH-extractable protein was
positively correlated to MAP (r = 0.46, p<0.01) and the removal of correlations with geochemical parameters had no
significant effect.







**Figure 4 Climate effects on gross protein depolymerization. (a),** Relationship between the natural logarithm of gross protein depolymerization and 2nd polynomial regression fit for cropland (adjusted r² = 0.455, p<0.001, n = 24), grassland (r² = 0.480, p<0.001, n = 28) and woodland (r² = 0.219, p<0.01, n = 48) soils. **(b),** Relationship between the natural logarithm of gross protein depolymerization rates and the ratio of mean annual precipitation over potential evapotranspiration (MAP:PET) and regression fit (y=log(x)) for cropland (adjusted r² = 0.330, p<0.01, n = 24), grassland (adjusted r² = 0.371, p<0.001, n = 28) and woodland (adjusted r² = 0.318, p<0.001, n = 48) soils. The vertical line indicates the transition from arid to humid climate conditions (MAP:PET = 0.65). **(c),** zero-order and partial correlations (Pearson's r) between climate variables (MAT and MAP) and organic N cycling (protein depolymerization rate, leucine-amino peptidase activity and Protein$_{NaOH}$) controlled for geochemical variables). Significant correlations are indicated by bold numbers. Significant changes of the correlation coefficients compared to the zero-order correlation are indicated by italic numbers.





## 3.3 Path analyses

The a priori model was constructed according to the hypothesis illustrated in Figure 1. After removing insignificant paths the model contained NaOH-extractable protein, soil pH, amorphous Fe and Al and MAP ($X^2$ = 2.49, p = 0.288; RMSEA = 0.048, SRMR = 0.023). The revised model explained 43% of the variance in gross depolymerization and 49% of the variance in NaOH-extractable protein. Protein depolymerization in mineral soils was highly dependent on NaOH-extractable protein. Soil pH had direct and indirect (via NaOH-extractable protein) negative effects on depolymerization rates (Figure 5). MAP and amorphous Fe and Al had positive effects on NaOH-extractable protein and thereby positive indirect effects on protein depolymerization. The total effects (direct effects + indirect effects) of the model parameters on protein depolymerization increased in the order amorphous Fe and Al<soil pH<MAP<NaOH-extractable protein. The model explained 49% of the variation of NaOH-extractable protein and 43% of the variation of protein depolymerization rates.

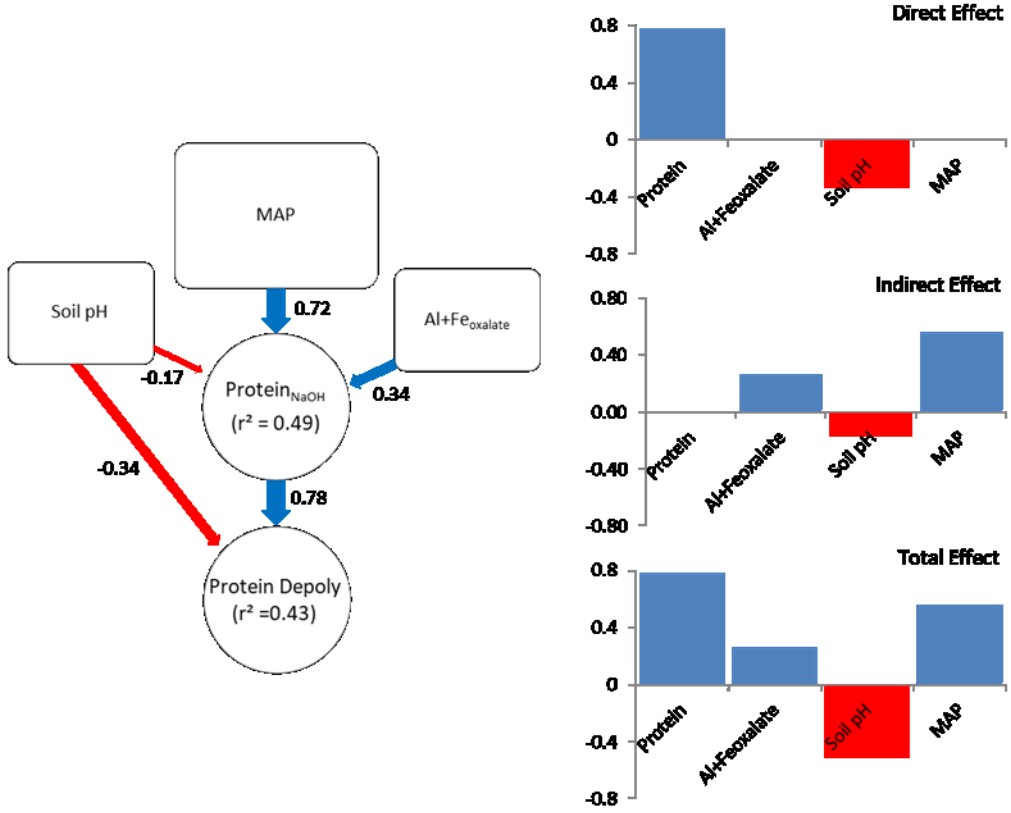

**Figure 5 Direct and indirect effects in gross protein depolymerization rates. Controls of Path analyses for gross protein depolymerization rates in mineral soils and coefficients for direct, indirect and total effects (n=91). Significant effects (p<0.05) are indicated by black arrows and effect sizes are indicated by line width. Numbers beside arrows indicate the standardized parameter estimates. Numbers within boxes indicate the variance explained by the model.**





## 4 Discussion

### 4.1 Land use and soil horizon effects on protein depolymerization

Across all land use types and soil horizons we found support for our hypothesis that gross protein depolymerization is controlled by N availability which increased with soil organic matter (SOM) contents from Mediterranean to temperate and
boreal ecosystems (Table S3). SOM has often been found to decline in cropland soils due to reduced root and litter input and due to tillage (Six and Jastrow, 2002). As a consequence NaOH-extractable protein and gross depolymerization were lowest in cropland soils compared to grassland and woodland soils across all biomes. We suggest that in the studied cropland soils depolymerization was likely limited by the availability of proteins, as SOM declines, though we found no differences between total N contents of the three land use types (Table S2). However, in cropland soils, depolymerization was strongly
correlated with root biomass which may indicate that easily available C compounds introduced into soils by root exudation and root turnover enable microorganisms to mine for organic N (Angst et al., 2018). In grassland soils we observed an increase of protein depolymerization rates with lower root C:N ratios, which was accompanied by a shift in microbial community composition from high fungi:bacteria ratios in N-limited grasslands towards lower fungal abundance in N-rich grasslands (Table S3). Decreases in fungi:bacteria ratios have also been reported with increases in N availability e.g. due to
fertilization (Grayston et al., 2001; De Vries et al., 2006). However, this pattern in protein depolymerization was not confirmed to be related to soil C:N, or to microbial C:N imbalance defined as the ratio of soil C:N over microbial C:N. In woodland soils, high SOM and NaOH-extractable protein contents did not translate into higher depolymerization rates compared to grassland soils. We suggest that the effect of SOM on depolymerization rates might be overridden by the large heterogeneity in forest edaphic properties e.g. soil pH ranging from 4.0 to 8.9. Gross depolymerization might be further
constrained by the lower peptidase activities found in woodland soils compared to grassland and cropland soils (Table S2) due to the typically lower soil pH (Sinsabaugh et al., 2008). Moreover, at acidic soil pH mobilization of $Al^{3+}$ might promote precipitation of proteins and of proteolytic enzymes, which restricts enzymatic cleavage (Scheel et al., 2008). Overall, land use had no effect on the response of depolymerization rates to soil properties and explained less than 5% of the variance, as shown by multiple linear regressions (Figure 3, Table S5), and might therefore be only a minor control of soil organic N
cycling across large scales. The positive effect of substrate availability on depolymerization rates was also confirmed by the high gross protein depolymerization rates observed in organic horizons in boreal and alpine biomes which exceeded those in the underlying mineral soils by 5-fold. In contrast to findings from Mooshammer at al. (2012) for decomposing litter, our data revealed no indication that resource C:N or microbial C:N imbalance affected protein depolymerization rates in organic soils. However, the increasing depolymerization rates with latitude though vegetation N limitation increases and the missing
effects of resource C:N ratio and microbial C:N imbalance indicate differential limitation of plants and soil microbes across large spatial scales as proposed by Capek et al. (2018).





## 4.2 Substrate limitation of protein depolymerization is controlled by organo-mineral interactions

Across all land use types NaOH-extractable protein and soil pH were the main predictors for gross protein depolymerization in mineral soils, indicating that soil properties that determine protein availability such as texture, mineral assemblage or soil
pH need to be considered when addressing controls of soil organic N cycling. Gross protein depolymerization was lower in soils developed on limestone than in soils developed on sediments or silicates, which is emphasized by the inverse relationship between depolymerization rates and soil pH (Figure 3). Moreover, depolymerization rates decreased with increasing clay content. Proteins can be adsorbed to clay surfaces by electrostatic interactions between positively charged amino acid side chains and siloxane surfaces of clay minerals (Staunton and Quiquampoix, 1994; Quiquampoix and
Ratcliffe, 1992). Sorption experiments in artificial soils showed that at neutral soil pH (>7) clay minerals are the main sorption sites for organic N (Pronk et al., 2013). This can be further enhanced by polyvalent cations as $Ca^{2+}$ or $Mg^{2+}$ which can bridge the negative charges of clay mineral surfaces and proteins (Cao et al., 2011; Lützow et al., 2006). Aside from the stabilization on mineral surfaces, high clay contents, as found in limestone soils, promote soil aggregation and thereby occlusion of organic matter and proteins rendering them inaccessible for enzymatic attack (Lützow et al., 2006). In contrast
Fe- and Al- oxyhydroxides, the main sorption sites for SOM at acidic pH, were positively correlated to gross depolymerization rates. SOM accumulation is usually higher in acid soils due to ligand exchange between protonated hydroxyl groups of Fe- and Al- minerals and carboxyl groups of organic molecules (Gu et al., 1994; Kleber et al., 2005; Kaiser and Guggenberger, 2000). In acidic soils, column experiments with embedded goethite revealed that sufficiently large amounts of stabilized C were re-dissolved by progressing percolation of dissolved OM and consequent exchange of adsorbed
compounds, indicating that stabilized compounds are available for microbial utilization (Leinemann et al., 2018). This is further supported by the strong positive correlation between NaOH-extractable protein and amorphous Fe- and Al-oxyhydroxides (Table S3), since NaOH mainly extracts loosely bound proteins (Wattel-Koekkoek et al., 2001). However, Fe- and Al oxyhydroxides remained as a significant parameter in linear models and path analyses and should therefore be considered as important predictor for the potential of a soil to retain and accumulate SOM, which promotes microbial live
and decomposition. However, the total N pool size was not significantly different between soils developed on the three bedrock types but NaOH-extractable protein increased on the order limestone<sediment<silicate. In contrast NaOH-extractable protein accounted for 4.4±1.7% of total N in sediment soils and for 6.4±3% in silicate soils. This could be either attributed to lower extraction efficiency of proteins with 0.5 M NaOH from clay minerals at high soil pH or to an increase of non-hydrolysable organic N. The studied limestone soils were characterized by higher amounts of crystalline iron ($Fe_{d-o}$),
namely hematite which forms almost irreversible interactions with SOM (Gu et al., 1995), even at high soil pH due to formation of coordination complexes between carboxyl groups and Fe atoms (Koutsoukos et al., 1983; Quiquampoix, 2000). The formation of strong complexes on crystalline Fe minerals is also supported by findings of Mikutta et al. (2010), who showed an increase of non-hydrolysable peptide-N with the proportion of crystalline Fe minerals across a soil chronosequence.





From linear regression and path analyses soil pH was revealed as the second most important predictor of gross depolymerization rates. Soil pH affects electrostatic interactions between mineral surfaces and proteins. Sorption of proteins on clay and Fe-mineral surfaces is usually highest close to the isoelectric point of a specific protein, when the net charge is zero and repulsion from charged surfaces is smallest. However, due to the complex nature of proteins including different functional groups and their tertiary structure isoelectric points range from pH 1 for pepsin to pH 11 for lysozyme, making

predictions for soil proteins at large impossible. Sorption of bovine and human serum albumin on montmorillonite peaked at pH ~5 whereas adsorption of cytochrome c or ribonuclease on hematite peaked at pH 8 to 10 (Khare et al., 2006; Koutsoukos et al., 1983; Quiquampoix and Ratcliffe, 1992).

         However, the negative effect of soil pH on gross depolymerization is in sharp contrast to the increase of peptidase activity with soil pH. To allow comparisons between enzyme activity and depolymerization rates enzyme activities were

measured in unbuffered soil slurries at natural soil pH compared to enzyme activities measured at same pH in acetate buffer (pH 5.2). Hence, unbuffered peptidase activities were highest in limestone soils close to the pH optima of proteolytic enzymes at about 8 (Sinsabaugh et al., 2008) (Figure S5). The lack of correlation between gross depolymerization and peptidase activity implies that gross protein depolymerization rates are rather substrate limited than enzyme limited and that differences in protein depolymerization rates between alkaline, neutral and acidic soils are due to changes in substrate

(protein) availability rather than due to changes in microbial community structure and enzymatic activity. Even when peptidase was measured at the same pH, potential peptidase activity was higher in limestone soils compared to sediment and silicate soils (Table S2, Figure S6), which implies enhanced microbial enzyme excretion in limestone soils in response to lower protein availability/concentration.

The generally low protein depolymerization rates in limestone soils are in accordance to our previous findings from soils

developed on limestone and silicate bedrock in Austria (Noll et al., 2019b), demonstrating that soil parent material pre-determines depolymerization rates on regional and continental scales. We assume that in limestone soils proteins are strongly stabilized on phyllosilicates and crystalline Fe-oxides or occluded within soil aggregates rendering them inaccessible for proteolytic attack. Soil microorganisms mitigate this N-limitation by mining for more recalcitrant SOM (Chen et al., 2014) as shown by the enhanced excretion of amino peptidase in limestone soils. With increasing amorphous Fe- and Al-

oxyhydroxides the sorption capacity of soils increases strongly due to their higher surface area (Kaiser and Guggenberger, 2003) facilitating accumulation of SOM. However, given the strong correlation between NaOH-extractable protein and amorphous Fe- and Al- oxyhydroxides, our data clearly shows that soil mineral assemblage is a crucial driver of soil organic N stocks and dynamics across large scales.

**4.3 Climate drives protein depolymerization by affecting mineral weathering and plant productivity**

Climate is a major control on mineral weathering and net primary productivity and thereby affects protein stabilization and input of fresh OM by plants. Across the studied climate transect gross protein depolymerization rates decreased with MAT and increased with MAP, respectively the climatic humidity index (MAP:PET). As demonstrated by the partial correlations,





part of the negative effect of MAT on depolymerization rates can be explained by concomitant changes in amorphous Al and Fe oxyhydroxides and soil pH which affect protein availability (Figure 4). The important role of soil geochemical properties on protein stabilization is underpinned by the even stronger effect of soil properties on the relation of MAT and NaOH-extractable protein (Figure 4). In the Mediterranean region limestone derived red soils are predominating. The so called "Terra Rossa" soils are characterized by high soil pH, high clay contents and higher amounts of crystalline Fe as well as a low $Fe_{oxalate}$:$Fe_{dithionite}$ ratio, caused by the preferential formation of the Fe-oxide hematite over the Fe-hydroxide goethite during the summer dry period (Yaalon, 1997). As described above, these specific soil properties might foster stabilization of proteins and thereby constrain gross depolymerization. Under more humid conditions soil pH drops due to leaching of base cations (e.g. $Ca^{2+}$) and more intensive chemical weathering causes higher amounts of charged mineral surfaces as amorphous Fe- and Al oxyhydroxides (Doetterl et al., 2015). This increase in soil acidification with latitude is further facilitated by predominance of silicate bedrocks in Northern Europe. Although MAP is an important driver of soil weathering and thereby affects soil pH and formation of charged mineral surfaces the positive effect of MAP on depolymerization rates and proteins was not significantly biased by soil properties (Figure 4). However, the weak effects of Fe and Al oxyhydroxides on the relation between protein depolymerization and MAP, or between NaOH-extractable protein and MAP might indicate the role of MAP in soil mineral formation during pedogenesis. Particularly in arid and sub-arid biomes precipitation determines plant net primary production (Yang et al., 2008; Del Grosso et al., 2008) and thereby the input of fresh organic matter into the soil. This might further explain the strong relationship between NaOH-extractable protein and MAP, as indicated by linear models and path analyses and is further supported by the proximate increase in depolymerization with the climatic humidity index (Figure 4). The logarithmic response implies that the limiting effect of MAP is stronger under sub-arid conditions, which is in accordance to findings showing that in water limited regions NPP is strongly controlled by MAP (Yang et al., 2008). Therefore, we conclude that, in sub arid regions in Southern Europe precipitation constrains plant biomass production and consequently OM input into soils. In contrast, our results reveal that the increase of gross depolymerization with MAT is biased by changes in soil parent material across the studied transect, while MAP likely controls net primary productivity and mineral weathering (Gislason et al., 2009; La Pierre et al., 2016). Both, partial correlations and path analyses support our hypothesis that climate is a rather indirect control on soil organic nitrogen cycling by its effects on soil geochemistry and soil organic matter accumulation.

Path analysis emphasized the important role of climate and bedrock as pre-determinants of OM stabilization and protein availability and suggested that MAP, soil pH and Fe- and Al- oxyhydroxides are indirect controls on gross depolymerization mediated by protein availability while soil pH and NaOH-extractable protein were direct controls on gross depolymerization. The indirect effect of MAP exceeded the direct effects of soil mineralogy and pH. However, NaOH-extractable protein was the main predictor of protein depolymerization rates. The negative direct effect of soil pH on depolymerization rates is explained by the low solubility of proteins at high soil pH (Franco and Pessôa Filho, 2011) which restricts diffusion throughout the soil matrix and limits accessibility of proteins to enzymatic attack. In contrast, the negative pH effect on NaOH-extractable protein is attributed to the accumulation of SOM at acidic soil pH and the enhanced





interactions with Fe- and Al- oxyhydroxides (Kaiser and Guggenberger, 2003; Gu et al., 1994). With increasing soil pH amino groups of proteins become de-protonated and thereby proteins become negatively charged which increases the repulsion from negatively charged mineral surfaces and decreases the adsorption to Fe-oxides and phyllosilicates (Cao et al.,

2011). Furthermore, soil pH, texture and mineral assemblage are drivers of microbial community composition and affect the availability of other nutrients like P or K (Fierer and Jackson, 2006; Lauber et al., 2008). Neither $Ca^{2+}$ nor clay was included in the final model, despite their important role in stabilizing soil organic matter (Lützow et al., 2006). We assume that the effects of $Ca^{2+}$ and clay are outweighed by effects of soil pH and MAP. Soil pH decreased from clay rich limestone soils to sediment soils and to more sandy silicate soils and thereby covaried with $Ca^{2+}$ and clay content, while MAP regulates

mineral dissolution and leaching of $Ca^{2+}$ (Gislason et al., 2009). Land use was also removed from the revised model which is in accordance to results from general linear models, showing that soil properties and climate variables explained a much higher percentage of the variance in gross protein depolymerization. Although path analyses provided an integrative model of controls driving gross protein depolymerization, it offered an incomplete picture. In this study we focused on the large scale patterns, which explained more than 40% of the variation in organic N cycling. However, regional or local effects, such

as topography, land use history/intensity or plant community composition, were not accessible with this data set, but are likely important controls on organic N cycling.

## 5 Conclusions

Our results highlight the important role of soil geochemistry when estimating microbial nutrient cycling on continental to global scales, and demonstrate that soil parent material and climate modulate the effects of land use on soil organic N

transformations. The amount of NaOH-extractable protein was identified as an important direct predictor of protein depolymerization rates. In contrast, peptidase activity was a poor proxy of protein depolymerization, but rather a proxy of enzyme production and of microbial C or N limitation. Since protein availability and thereby depolymerization is strongly constrained by soil mineral interactions, shifts in climate (precipitation regime) and associated alterations in soil weathering should be considered as drivers of ecosystem N availability with repercussions on ecosystem C cycle processes. This also

needs to be validated in large-scale coupled biogeochemical climate models to help predict and mitigate global change effects.

## Data availability

All data and codes presented in this paper are available upon reasonable request.





## Author contribution

LN wrote the paper, conducted fieldwork and laboratory work, analyzed and interpreted the data. SZ, QZ and YH conducted laboratory work, analyzed the data and edited the paper. FH analyzed the data and edited the paper. WW designed the study, interpreted the data and edited the paper.

## Competing interests

The authors declare that they have no competing interests.

## Acknowledgements

We thank Theresa Böckle, Daniel Wasner, Vsevolods Girsovics and Rebecca Lieske for soil sampling and assistance in the lab. We would like to thank Jukka Pumpanen for providing soil samples from the Värriö District Nature Reserve. This project was funded by the Austrian Science Fund (FWF, Project P-28037-B22).

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
