# Peer review of "Climate and geology overwrite land use effects on soil organic nitrogen cycling on a continental scale."

_Biogeosciences, 2022_

## Author Response (AR1)

1. **Response to reviewer comments**

    1.1. **Reviewer Comment #1**

    **This study studied the factors of land use and bedrock on protein depolymerization across a 4000-km transect in Europe, and highlighted the important role of climate and soil properties on N cycling at large scales. The sampling scheme is attractive, and many interesting biochemical indicators are measured. This study stated N cycling was controlled by substrate availability. This conclusion is not novel, actually which has been widely acknowledged for a long time. Generally, I think this paper fits the scope of this Journal, however, more should be added to highlight the novelty of this study. Some conclusions are confusing, the author had clarified the importance of substrate availability, which was related with OM input or vegetation, however, it also declared that the land use effect was insignificant. Those ideas are not consistent. It's better to clarify them. Besides, the writing should be improved. For instance, some paragraphs lack the key points, and some sentences are confusing or too long to understand. See the details as follows.**

    We thank the reviewer for the critical comments on our manuscript.

    Minor comments to RC1 suggestions/criticisms: Our conclusion that N cycling is controlled by substrate availability is not our conclusion, we concluded that substrate (protein) availability drives gross rates (in situ rates) of protein depolymerization in soils across a continental transect. This is different from pouring some substrate on a soil and finding changes in net nitrification or net N mineralization, as we here measured gross process rates. Moreover, we here refer to organic N and not inorganic N cycling. We are aware that protein addition to soils increased proteolytic activities in arctic soils (Weintraub & Schimel, 2005), but this also does not refer to a gross process but rather to potential activities, which we here demonstrated not to be the driver of in situ depolymerization rates. Our MS here is only the second paper showing substrate limitation of the depolymerization processes in soils, after Noll et al. 2019 at the regional scale, but here at a large scale. This is also the major novelty of this study. In the revised MS we will improve this. We will also improve the text to highlight the key points in each paragraph (upfront or at the end). Too long or confusing sentences will be cut and rewritten.

    Below are itemized replies to the referee comments. The line numbers refer to those of the original manuscript.

    **Line 40: The key point of this paragraph is not clear, clarify it. And since land use is your main sampling scheme, you can introduce more about it.**

    The aim of the study was to disentangle the large scale controls on gross protein depolymerization rates and organic nitrogen cycling. Since land use is one determinant of soil functions and soil microbial community structure and function, different land uses were included in the study design. However, land use certainly was not determining the study in terms of main sampling scheme. We here nested land use in large scale climatic and geological controls along this continental transect, but searching for sites distributed across whole Europe where we found at least two (at best: three) land uses in close vicinity (max. few 100s meter). Moreover, effects of land use are expected to be more prominent at a smaller regional to local scale and exact land use data were not accessible for this data set. On the continental scale climate and bedrock considered the main drivers of organic N cycling as demonstrated by our results, though this has been rarely studied in such a systematic way. Therefore, land use was an important factor in our sampling scheme, yet not the one and only, and likely being strongly overprinted

by large scale changes in climate and geology. We will clarify this in the introduction and discussion section of the revised manuscript. We also will add 1-2 more sentences on land use effects on soil organic N dynamics.

**Line 50. This reference was in contrast with your contents, discuss it later.**

The reference is not in contrast with our results. In the study by Mooshammer et al. (2012) the effects of elemental stochiometry (C:N:P) on gross protein depolymerization rates were studied in decomposing beech litter. In contrast, in more decomposed organic horizons, processes of organic matter stabilization are increasingly important and might mask the stoichiometric effects reported from litter studies. We will further clarify this in the introduction and discussion section of the revised manuscript.

**Line 64, add reference.**

We will add appropriate references such as Lauber et al. for global drivers of microbial community structure, and global meta-analyses on net and gross N mineralization and nitrification (published by Christoph Müller's group), and on soil and microbial C:N:P stoichiometries on the global level. We rephrased the sentence as follows

*In conclusion, land use, bedrock and biogeographic region are likely key controls on soil nutrient status and edaphic properties and affect microbial community structure, substrate availability and microbial N and C demands as shown in Figure 1.*

**Line 96, what's the depth of the 'organic layers'?**

Organic layers were sampled down to the mineral soil. The depth of the individual organic horizons varied strongly from 2 cm to > 15 cm (max. 30 cm). This will be added to the revised MS.

**Line 180: This sentence is confusing, please rephrase it.**
We rephrased the sentence as follows:

*For statistical analyses of single variables, mineral soils were grouped by bedrock (limestone, sediments, silicates) or by land use (cropland, grassland, woodlands).*

**Line 340: Simplify the sentences**

We rephrased the sentence as follows:

*However, though vegetation N limitation increases with latitude (Kang et al, 2010) we showed here that depolymerization rates increased with latitude, indicating increasing labile organic N provisioning to microbes and plants at higher latitudes under lab conditions. This highlights differential element viz. nutrient limitation of plants and soil microbes across large spatial scales as proposed by Capek et al. (2018). This is also supported by the missing effects of resource C:N ratios and microbial C:N imbalances on depolymerization rates.*

**Lin 320-335: It's confusing in 4.1. 'land use had no effect on the response of depolymerization rates', however, above discussion was talking about the differences in different land use, and even attributed the difference to soil pH.**

The described land use effects were significant for individual sampling sites. However, in the overall statistical analyses the effect of land use explained only 5% of the variability. We will clarify this in the discussion section of the revised manuscript.

**Line 343: pH was the main predictor in results, but the contribution of texture, mineral assemblage, how are they related to N cycling﹖**

In line 343 we stated that "Across all land use types NaOH-extractable protein and soil pH were the main predictors for gross protein depolymerization in mineral soils, indicating that soil properties that determine protein availability such as texture, mineral assemblage or soil pH need to be considered when addressing controls of soil organic N cycling.". This does not imply that as RC1 mentions "pH was the main predictor", and then asks how texture and mineral assemblage are related to N cycling". Soil pH mirrors the strength of Ca-bridging of negatively charged ligands (as protein-carboxylates) to negatively charged soil particles (clays), but also the weathering status of soils, which comes with the formation of secondary clays and Fe/Al oxyhydroxides. On the other hand, soil texture and mineral assemblage are also affected by geology (aside of weathering) and directly affect the amount of organic N bound and at the same time the binding strength of this interaction. Gross protein depolymerization rates were negatively correlated to clay content, indicating that the availability of proteins decreases with increasing clay content. Sorption experiments in artificial soils showed that at neutral soil pH (>7) clay minerals are the main sorption sites for organic N (Pronk et al., 2013). Aside from the stabilization on mineral surfaces, high clay contents, as found in limestone soils, promote soil aggregation and thereby the occlusion of organic matter and proteins rendering them inaccessible for enzymatic attack (Lützow et al., 2006). We will clarify this in section 4.1.

**Line 360: I don't think this indicate that 'stabilized compounds are available for microbial utilization'**

We rephrased the sentence as follows:

*In acidic soils, column experiments with embedded goethite revealed that sufficiently large amounts of stabilized C were re-dissolved by progressing percolation of dissolved OM and consequent subsequent exchange of adsorbed compounds. The re-dissolved compounds are thus available for microbial utilization (Leinemann et al., 2018).*

**Line 365: Add reference**

"Fe- and Al oxyhydroxides remained as a significant parameter in linear models and path analyses and should therefore be considered as important predictor for the potential of a soil to retain and accumulate SOM (Moni et al, 2007; Fang et al., 2019), high SOM promoting microbial biomass and activity (Xu et al., 2013; Hartman & Richardson, 2013)."

We will add appropriate references here, such as:

Moni, C., Chabbi, A., Nunan, N., Rumpel, C., & Chenu, C. (2007, December). Do iron and aluminium oxides stabilise organic matter in soil? A multi-scale statistical analysis, from field to horizon. In AGU Fall Meeting Abstracts (Vol. 2007, pp. B11G-04).

Fang, K., Qin, S., Chen, L., Zhang, Q., & Yang, Y. (2019). Al/Fe mineral controls on soil organic carbon stock across Tibetan alpine grasslands. *Journal of Geophysical Research: Biogeosciences*, *124*(2), 247-259.

Xu, X., Thornton, P. E., & Post, W. M. (2013). A global analysis of soil microbial biomass carbon, nitrogen and phosphorus in terrestrial ecosystems. Global Ecology and Biogeography, 22(6), 737-749.

Hartman, W. H., & Richardson, C. J. (2013). Differential nutrient limitation of soil microbial biomass and metabolic quotients (q CO2): is there a biological stoichiometry of soil microbes?. PloS one, 8(3), e57127.

**Line 376-378: this sentence is confusing, please simplify it**

We rephrased the sentence as follows:

*Sorption of proteins on clay and Fe-mineral surfaces is usually highest close to the isoelectric point of a specific protein.*

**Line 390: add support for this opinion**

In these paragraph we highlight one major outcome of this large scale study, which we consider robust based on multiple data lines presented in this MS, i.e. "The lack of correlation between gross depolymerization and peptidase activity implies that gross protein depolymerization rates are rather substrate limited than enzyme limited. Differences in protein depolymerization rates between alkaline, neutral and acidic soils are due to changes in substrate (protein) availability rather than due to changes in microbial community structure and in enzymatic activity.". The whole MS builds towards this "conclusion", we therefore consider it unnecessary to add further (?) support here.

**Line 405-420: Do you mean that the climate factors influence soil pH and then regulate the depolymerization rates? However, your data doesn't seem to support this, please explain it.**

We concluded that climate factors drive chemical weathering of soil minerals. These changes are accompanied by the mobilization and hydrological losses of base cations such as K, Ca and Mg and consequently a drop of soil pH. However, our data show that depolymerization rates are not controlled by soil pH in the first place. Our data rather support that the observed effect of climate factors on depolymerization rates is explained by changes in soil chemical weathering and more specifically the formation of specific minerals. We will clarify this in the revised manuscript.

**Line 450: can you compared the contributions in the combined model including land use, soil properties, climate together?**

As shown in Fig. 4 a-e land use did not affect the response of gross depolymerization rates to changes in soil properties (this would be seen in differences in the slopes of these relationships between croplands, grasslands and woody vegetation), but contributed to differences in soil pH and NAOH-extractable protein between land uses, as shown by different intercepts of the linear regression models (Fig. 4f). The regression model on climate factors and land use (Fig. S5) also shows that land use is only a minor driver of depolymerization. Therefore, land use was not included in the final SEM and the SEM only explains the large scale soil physicochemical controls on protein depolymerization. Land use effects might become more dominant on regional scales, where effects of climate are smaller.

**Line 462: It's confusing, 'peptidase activity is a proxy of microbial N or N limitation'?**

This refers to the work by Allison, Sinsabaugh, Weintraub and others, and the theory of enzyme allocation. According to Allison et al. (Allison et al.,2010) "Extracellular enzymes allow microbes … to acquire resources from complex molecules, …. We examine the hypothesis that extracellular enzyme producers are under evolutionary pressure to minimize the cost:benefit ratio of enzyme production. Consistent with this prediction, enzyme producers generally allocate more resources to enzymes that target limiting nutrients." According to this peptidase activity can be used as a proxy for microbial N limitation.

We rewrote this sentence as follows:

*The amount of NaOH-extractable protein was here identified as the most important direct predictor of protein depolymerization rates, while peptidase activity was a poor predictor of protein depolymerization, but rather reflects a proxy of microbial N limitation according to enzyme allocation theory (Allison et al., 2010.*

**Fig 5: 'black arrows' are missing in the figure. And how to identify the direct and indirect effects?**

The figure caption describes the black/white plot of Fig. 5. In the manuscript we showed the colored plot with red and blue arrows. We will correct the figure captions in the revised manuscript.
The indirect effects are effects of one parameter on depolymerization rates mediated by another parameter. For example mean annual precipitation has an indirect effect on depolymerization rates via NaOH-extractable protein.

**References**

Weintraub, M. N., & Schimel, J. P. (2005). Seasonal protein dynamics in Alaskan arctic tundra soils. Soil Biology and Biochemistry, 37(8), 1469-1475.

Kang, H., Xin, Z., Berg, B., Burgess, P. J., Liu, Q., Liu, Z., ... & Liu, C. (2010). Global pattern of leaf litter nitrogen and phosphorus in woody plants. Annals of forest science, 67(8), 811.

Capek, P. T., Manzoni, S., Kastovska, E., Wild, B., Diakova, K., Barta, J., Schnecker, J., Blasi, C., Martikainen, P. J., Alves, R. J. E., Guggenberger, G., Gentsch, N., Hugelius, G., Palmtag, J., Mikutta, R., Shibistova, O., Urich, T., Schleper, C., Richter, A., and Santruckova, H.: A plant-microbe interaction framework explaining nutrient effects on primary production, Nature Ecology & Evolution, 2, 1588-1596, 10.1038/s41559-018-0662-8, 2018.

Pronk, G. J., Heister, K., and Kögel-Knabner, I.: Is turnover and development of organic matter controlled by mineral composition?, Soil Biology and Biochemistry, 67, 235-244, 2013.

Lützow, M. v., Kögel-Knabner, I., Ekschmitt, K., Matzner, E., Guggenberger, G., Marschner, B., and Flessa, H.: Stabilization of organic matter in temperate soils: mechanisms and their relevance under different soil conditions–a review, European Journal of Soil Science, 57, 426-445, 2006.

Allison, S. D., Weintraub, M. N., Gartner, T. B., & Waldrop, M. P. (2010). Evolutionary-economic principles as regulators of soil enzyme production and ecosystem function. In Soil enzymology (pp. 229-243). Springer, Berlin, Heidelberg.

**1.2. Reviewer Comment #2**

In this work, Noll et al. examine controls on protein depolymerization rates, a known key step in the production of LMWON compounds that can be used by microbes and (sometimes) plants. They find that substrate availability is a key control on depolymerization, and in turn they identify soil pH, MAP, and Al/Fe oxyhydroxides as key controls on substrate availability. The study is a wide-ranging longitudinal soil survey across Europe, from the Mediterranean to the Barents Sea. They find that land use has a negligible effect on substrate availability and depolymerization rates, somewhat surprisingly. They reached these conclusions through a combination of anova/linear regression approaches and structural equation models. The observational breadth and depth of this study is quite impressive. Fourty three sites across Europe were sampled, and exhaustive chemical and biological analyses were performed on the soils. The key measurements are well-supported from a theoretical standpoint. This seems like a monumental effort that was well-planned and carefully executed.

The manuscript definitely needs some honing to make the central story stand out more, though. I also encourage the authors to rethink their statistical approaches; I'm not asking them to redo all of their analyses, but I think a shift in emphasis toward highlighting the analyses that deal with the highly correlated nature of the predictors, is warranted. I honestly really struggled through reading this paper. There are SO many measurements taken, and the results are presented in such exhaustive detail, that I found myself losing the thread often and wondering why data was being presented / what the main thrust of the argument was. I very strongly encourage the authors to revisit all of the topic sentences for each paragraph and make sure that the conclusions that should be drawn from a paragraph are clearly stated up-front. I also encourage the authors to think very carefully about what data are actually central to the story, and to shunt a lot of their other results to the supplement. Regarding the analyses, the authors are dealing with a ton of very highly correlated predictor variables, an issue which they recognize. They lead their results and discussion, though, with exhaustive treatment of single-variable ANOVAs (over sixty ANOVAS) which do not do justice to this rich but highly correlated predictor dataset. The authors have a very nice conceptual model (Fig 1) that is very nicely examined through an SEM. I think that this should be the centerpiece of the story! Some ordination approaches also would make more sense to me in terms of understanding the highly correlated nature of the data, rather than picking apart tables of bivariate correlation coefficients.

The paper also needs to be brought into compliance with EGU's data policy.

This is a nice body of work, and some careful editing will go a long way to making the story in this paper shine. (I'm also sorry the authors have waited 5 months and had many declined review requests. Frustrating!)

Best wishes,
Richard Marinos, U @ Buffalo

Dear Richard,

Thank you for your very positive response.

We highly appreciate your thorough review of the manuscript and your critical comments. We agree that the complexity of the data set and the high number of measurements and analyses make it difficult to stay focused on the central theme. We do fully understand this criticism and will thoroughly revisit our manuscript, better highlighting the central story, building this around the SEM in the Discussion section, removing more side data from Results to the Supplement, and trying to further reduce the data set using multivariate approaches. In the revised MS version we will shift even more of the "primary data" into the Supplement, though we need to mention that all single parameter analyses not central to the story were already in the Supplement in the first version. Anyway, the results section will be streamlined and important side results put in a Supplementary results section. Moreover, in all paragraphs we will take care to put the "take home message" upfront and then explain our reasoning. Certainly some parameters are highly correlated (e.g. positively soil microbial biomass C and N, negatively sand and clay, soil pH and exchangeable Ca and base saturation, etc.), while others are not or not necessarily (not talking about strong but spurious correlations). In our previous statistical analyses, in PCAs and CCAs and particularly in the SEM analysis we were fully aware of this highly correlated nature of specific soil properties and accounted for this by omitting highly co-varying parameters. This will be more clearly stated in the revised manuscript.

We also agree that the results sections focused too strongly on the single-variable ANOVAS, which are only used to support the results of our multivariate modelling but provide only limited information by their own. We will shorten the results section and transfer part of the results to the supplement.

You also suggest showing some ordination approaches. During the course of analyses we also tested some ordination approaches, mainly PCAs. But in our opinion the multivariate statistical results provided only limited insight. However, we will re-evaluate those analyses and examine if the multivariate analyses would help to increase the understanding of the highly correlated nature of the data.

Please find below our replies to the itemized referee comments. The line numbers refer to those of the original manuscript.

**Line items:**
**180 - I don't understand what the other factor, besides land use type, is in these models. More broadly, this analysis scheme doesn't make too much sense to me... your conclusions are that bedrock type is a key driver of depoly rates, and land use type is not. But bedrock type was only subject to a 1-way anova, while land use is subject to a 2-way anova which controls for bedrock type/climate. Why, for example, was the effect of bedrock type not analyzed with a 2-way ANOVA that controlled for the effects of land use? Given relatively low sample #s, it is unsurprising that there is not enough statistical power to detect an effect of land use type in a 2-way ANOVA when controlling for bedrock/climate/soil type, but the bedrock type analysis was not subject to the same dilution of statistical power, so it seems to me that the conclusions are drawn from incommensurable statistical approaches.**

For the analyses of land use effects we run the two-way ANOVA for the main effects of "land use" and "site" (with no interaction – see below), where the factor "site" controlled for any difference in climate,

geological substrate and soil type across the sites. We did this since at any site we had only one composite sample analyzed per land use at this site (no replication within land use x site) and therefore the single observations were not independent. We sampled the three land use types at each site in close vicinity and therefore only minor differences in bedrock, climate and soil properties were expected within site. But we did not control the "land use" ANOVA for bedrock/climate/soil properties in particular; for this we would have needed at least 10-fold larger numbers of samples/sites. However, we are aware, that land use effects are difficult to compare to the rather large scale controls, i.e. climate and soil properties. The chosen sampling scheme was more suitable to investigate large scale controls, where land use effects might be more predominant on regional to local scales. We will clarify this in the revised manuscript.

**205- Are the +/- numbers one standard error of the mean? Confidence interval? Please state at the first instance.**

The +/- numbers refer to one standard error of the mean. We will indicate this at the first instance in the revised manuscript.

**Figure 4 - Is there really a clear enough justification to use polynomial regression?**

The visual fit improved strongly when using a polynomial regression compared to a linear regression. We will provide further justification for the used model in the revised manuscript.

**360 - I have a hard time wrapping my head around how Fe and Al oxyhydroxides can simultaneously increase SOM stabilization AND increase SON availability.**

First, this conclusion is based on the fact that soils with larger amounts of minerals with very high specific surface areas such as Fe/Al oxyhydroxides and finer texture potentially store more SOM than soils with coarser texture. Therefore the overall organic N pool size is expected to be larger in fine textured soils and in soils high in Fe/Al oxyhydroxides. Second, the strength of the binding interaction between Fe/Al oxyhydroxides and SOM, and more specifically with organic N including proteins, is high, and higher than with typical clay minerals (Newcomb et al, 2017). Overall, this means that soils rich in Fe/Al oxyhydroxides contain larger pools of proteolytic substrates (organic N and proteins), but these substrates are more strongly bound and therefore less accessible. The net effect of these adverse interactions is currently unknown; therefore this study is among the first to show a net positive effect on the in situ rates of depolymerization of high molecular weight -ON substrates. Moreover, our conclusion is supported by the findings of Leinemann et al. (2018), showing that stabilized C was re-dissolved by progressing percolation of OM, indicating that organic compounds can be easily exchanged from e.g. goethite. Hence a higher amount of Fe/Al – oxyhydroxides might corresponds to a larger fraction of weakly bound organic N, which is continuously re-dissolved and becomes thereby available for microbial utilization. We will further clarify this in the revised manuscript.

We rephrased the sentence as follows:

*In acidic soils, column experiments with embedded goethite revealed that sufficiently large amounts of stabilized C were re-dissolved by progressing percolation of dissolved OM and consequent exchange of adsorbed compounds. The re-dissolved compounds thus become available for microbial utilization (Leinemann et al., 2018).*

**405 - "As demonstrated by partial correlations..." This statement makes an assumption that Al/Fe oxyhydroxides and pH are the TRUE controls, and MAT is just a latent predictor, which I don't think has been fully justified.**

In the statement we concluded that "*part of the negative effect of MAT on depolymerization rates can be explained by concomitant changes in amorphous Al and Fe oxyhydroxides and soil pH*". Increasing temperatures would rather be expected to directly positively affect soil enzyme activities and to promote substrate and enzyme diffusion for enzyme-substrate encounter and to trigger catalytic action, instead of directly negatively affecting depolymerization rates. The MAT-depolymerization relationship therefore must be indirect. The partial correlations are one way to depict direct and indirect effects on, and primary and secondary drivers of biogeochemical processes. They showed a significant decrease of the correlation coefficient of mean annual temperature and depolymerization by removing the effects of soil mineral Fe and Al contents. The decrease of the correlation coefficient by removing effects of soil pH was not significant. However, after removing effects of soil pH and Al/Fe oxyhydroxides the effect of mean annual temperature on depolymerization rates was still significant. We will clarify this in the revised manuscript MAT likely controls depolymerization indirectly by multiple effects on vegetation, soil weathering, microbial community structure etc..

**468 - Biogeosciences requires data to be published in a FAIR repository, or else have the reasons for data remaining unpublished be clearly explained. This is statement is not in compliance with those requirements. I also encourage the authors to archive their code.**

The presented data are part of a larger project. At the time of submission not all data from the project were published yet. However, we will deposit the here cited data in a data repository after acceptance of the manuscript (DRYAD, https://datadryad.org/stash).

**References**

Newcomb, C.J., Qafoku, N.P., Grate, J.W. *et al.* Developing a molecular picture of soil organic matter–mineral interactions by quantifying organo–mineral binding. *Nat Commun* **8**, 396 (2017). https://doi.org/10.1038/s41467-017-00407-9

Leinemann, T., Preusser, S., Mikutta, R., Kalbitz, K., Cerli, C., Höschen, C., Mueller, C. W., Kandeler, E., and Guggenberger, G.: Multiple exchange processes on mineral surfaces control the transport of dissolved organic matter through soil profiles, Soil Biology and Biochemistry, 118, 79-90, https://doi.org/10.1016/j.soilbio.2017.12.006, 2018.

2. **List of relevant changes**

   2.1. **General changes**

We shortened the results section 3.1 as follows:

*Protein depolymerization rates were strongly related to soil physicochemical properties like soil pH, amorphous Fe and Al minerals ($Fe_{oxalate}$, $Al_{oxalate}$) as well as to soil organic matter ($C_{org}$, total N), NaOH-extractable protein and microbial biomass ($C_{mic}$, PLFA) (Figure 3, Table S3). NaOH-extractable protein content increased with SOC, soil TN, root biomass and amorphous Fe- and Al-(hydr)oxides (Table S3, Figure 3). Soil pH was negatively correlated with gross depolymerization and NaOH-extractable protein, but positively to peptidase activity (Figure 3). However, across all sites as well as within subgroups we found no significant (putatively positive) correlation between aminopeptidase activity, a wide spread soil proteolytic enzyme, and protein depolymerization rates (Figure S5). In order to further examine the potential edaphic controls on gross protein depolymerization in mineral soils as well as interaction effects with land use we used multiple linear regression analyses. In the most parsimonious model NaOH-extractable protein explained 37% of the variance, emphasizing the prominent role of substrate availability controlling depolymerization rates (Figure 2). Land use did not interact with specific edaphic properties, and linear mixed effect models with land use as random factor confirmed the suggested main controls on depolymerization rates, i.e. protein availability and soil pH (Table S4).*

*Climate effects on depolymerization rates were analyzed by linear regression analyses including climate parameters, land use and interaction effects. We found significant effects of mean annual temperature (MAT) and mean annual precipitation (MAP) and of their interaction (MAP:MAT) (Table S5). Land use had no significant effect on the climate response of protein depolymerization, as shown by similar negative correlations between depolymerization and MAT in all three land use types (Figure 4). The model explained about 42% of the variance. Although the climatic humidity index (MAP:PET), expressed as MAP over potential evapotranspiration (PET), was not included in the most parsimonious model, the strong logarithmic increase of depolymerization rates with climatic humidity ($r^2=0.632$, $p<0.001$) across all sites and land use types was striking (Figure 4). The most parsimonious linear mixed effect model included land use as random factor and showed a strong negative effect of MAT and a positive effect of MAP. The model explained about 47% of the variance in protein depolymerization.*

We shortened section 4.1 as follows:

*Our results revealed, that land use, which is an important driver of SOM contents and soil microbial community composition (Lauber et al., 2008; Jangid et al., 2008) and consequently  of the set of excreted proteolytic enzymes (Lauber et al., 2008; Jangid et al., 2008), might only beexert a minor control ofn soil organic N cycling acrossat large spatial scales. Though effects were significant for individual sampling sites (Table S2), land use had no significant effect on the response of protein depolymerization rates to soil properties, explaining less than 5 % of the total variation in multiple linear regression models (Figure 3, Table S5). This demonstrates that the same drivers operated on protein depolymerization in croplands, grasslands and woodlands, and triggering the same directional and strength of response across land uses. Effects of land use were therefore likely strongly overprinted by large scale changes in climate and geology, since in the applied sampling scheme the factor land use was nested in large scale climatic and geological controls across a continental transect. Effects of land use might be more prominent at a smaller regional to local scale (Noll et al. 2019b), which was, however, not accessible with this data set.*

*At the continental level, gross protein depolymerization rates increased with rising soil organic matter (SOM) contents, from Mediterranean to temperate and boreal ecosystems. Though vegetation N limitation increases with latitude (Kang et al, 2010, Du et al., 2020; Augusto et al., 2017)REF, the rising depolymerization rates with latitude, indicate increasing labile organic N provisioning to microbes and plants at higher latitudes under lab conditions. This positive effect of substrate availability on depolymerization rates was further confirmed by high gross protein depolymerization rates observed in organic horizons in boreal and alpine biomes, which significantly exceeded those in the underlying mineral soils (Table S2). However, in contrast to findings of Mooshammer at al. (2012) for decomposing litter, our data revealed no indication that resource C:N or microbial C:N imbalances affected protein depolymerization rates in organic soils and thereby highlights the differential element viz. nutrient limitation of plants and soil microbes across large spatial scales as proposed by Capek et al. (2018).*

**2.2. Reviewer comment #1**

**Line 50. This reference was in contrast with your contents, discuss it later.**

The reference is not in contrast with our results. In the study by Mooshammer et al. (2012) the effects of elemental stochiometry (C:N:P) on gross protein depolymerization rates were studied in decomposing beech litter. In contrast, in more decomposed organic horizons, processes of organic matter stabilization are increasingly important and might mask the stoichiometric effects reported from litter studies. We will further clarify this in the introduction and discussion section of the revised manuscript.

**Line 64, add reference.**

We added the following references: Lauber et al, 2008; Lauber at al., 2009; Xu et al., 2013, Elrys at al., 2021

**Line 96, what's the depth of the 'organic layers'?**

We added the following sentence: *The depth of the individual organic horizons varied from 2 to 30 cm.*

**Line 180: This sentence is confusing, please rephrase it.**

We rephrased the sentence as follows:

*For statistical analyses of single variables, mineral soils were grouped by bedrock (limestone, sediments, silicates) or by land use (cropland, grassland, woodlands).*

**Line 340: Simplify the sentences**

We rephrased the paragraph as follows:

*At the continental level, gross protein depolymerization rates increased with rising soil organic matter (SOM) contents, from Mediterranean to temperate and boreal ecosystems. Though vegetation N limitation increases with latitude (Kang et al, 2010, Du et al., 2020; Augusto et al., 2017)REF, the rising depolymerization rates with latitude, indicate increasing labile organic N provisioning to microbes and plants at higher latitudes under lab conditions. This positive effect of substrate availability on depolymerization rates was further confirmed by high gross protein depolymerization rates observed in*

*organic horizons in boreal and alpine biomes, which significantly exceeded those in the underlying mineral soils (Table S2). However, in contrast to findings of Mooshammer at al. (2012) for decomposing litter, our data revealed no indication that resource C:N or microbial C:N imbalances affected protein depolymerization rates in organic soils and thereby highlights the differential element viz. nutrient limitation of plants and soil microbes across large spatial scales as proposed by Capek et al. (2018).*

**Lin 320-335: It's confusing in 4.1. 'land use had no effect on the response of depolymerization rates', however, above discussion was talking about the differences in different land use, and even attributed the difference to soil pH.**

We rephrased the paragraph as follows:

*Our results revealed, that land use, which is an important driver of SOM contents and soil microbial community composition (Lauber et al., 2008; Jangid et al., 2008) and consequently of the set of excreted proteolytic enzymes (Lauber et al., 2008; Jangid et al., 2008), might only beexert a minor control ofn soil organic N cycling acrossat large spatial scales. Though effects were significant for individual sampling sites (Table S2), land use had no significant effect on the response of protein depolymerization rates to soil properties, explaining less than 5 % of the total variation in multiple linear regression models (Figure 3, Table S5). This demonstrates that the same drivers operated on protein depolymerization in croplands, grasslands and woodlands, and triggering the same directional and strength of response across land uses. Effects of land use were therefore likely strongly overprinted by large scale changes in climate and geology, since in the applied sampling scheme the factor land use was nested in large scale climatic and geological controls across a continental transect. Effects of land use might be more prominent at a smaller regional to local scale (Noll et al. 2019b), which was, however, not accessible with this data set.*

**Line 343: pH was the main predictor in results, but the contribution of texture, mineral assemblage, how are they related to N cyclingｼ¼Ÿ**

We rephrased the paragraph as follows:

*Therefore the overall organic N pool size is expected to be larger in fine textured soils and in soils high in Fe- and Al- oxyhydroxides. Moreover, the strength of the binding interaction between Fe- and Al- oxyhydroxides and SOM, and more specifically with organic N including proteins, is higher by 50% than with typical clay minerals (Newcomb et al, 2017). Consequently soils rich in Fe- and Al- oxyhydroxides contain larger pools of proteolytic substrates (organic N and proteins), but these substrates arecan be more strongly bound and therefore be less accessible for microbial utilization.In acidic soils, column experiments with embedded goethite revealed that sufficiently large amounts of stabilized C were re-dissolved by progressing percolation of dissolved OM and consequent exchange of adsorbed compounds, indicating that stabilized compounds are available for microbial utilization (Leinemann et al., 2018) However, column experiments with embedded goethite in acidic soils revealed that sufficiently large amounts of stabilized COM can be re-dissolved by progressing percolation of dissolved OM and the consequent subsequent exchange ofwith adsorbed compounds such as peptides (Leinemann et al., 2018), which thereby become available for enzymatic attack and/or microbial utilization. Hence a higher amount of Fe- and Al – oxyhydroxides might corresponds to a larger fraction of weakly bound organic N, which is continuously re-dissolved and becomes thereby available for microbial utilization... This The bioavailability of oxide-bound organic N is further supported by the strong positive correlation between NaOH-extractable protein and amorphous Fe- and Al- oxyhydroxides (Table S3), since NaOH mainly extracts loosely bound proteins (Wattel-Koekkoek et al., 2001).*

We added the following sentence:

*Soil pH mirrors the strength of $Ca^{2+}$-bridging of negatively charged ligands (as protein-carboxylates) to negatively charged soil particles (clays), but also the weathering status of soils, which comes with the formation of secondary clays and Fe- and Al-oxyhydroxides.*

**Line 360: I don't think this indicate that 'stabilized compounds are available for microbial utilization'**

See our changes above.

**Line 365: Add reference**

We added the following references:

Moni, C., Chabbi, A., Nunan, N., Rumpel, C., & Chenu, C. (2007, December). Do iron and aluminium oxides stabilise organic matter in soil? A multi-scale statistical analysis, from field to horizon. In AGU Fall Meeting Abstracts (Vol. 2007, pp. B11G-04).

Fang, K., Qin, S., Chen, L., Zhang, Q., & Yang, Y. (2019). Al/Fe mineral controls on soil organic carbon stock across Tibetan alpine grasslands. *Journal of Geophysical Research: Biogeosciences*, *124*(2), 247-259.

Xu, X., Thornton, P. E., & Post, W. M. (2013). A global analysis of soil microbial biomass carbon, nitrogen and phosphorus in terrestrial ecosystems. Global Ecology and Biogeography, 22(6), 737-749.

Hartman, W. H., & Richardson, C. J. (2013). Differential nutrient limitation of soil microbial biomass and metabolic quotients (q CO2): is there a biological stoichiometry of soil microbes?. PloS one, 8(3), e57127.

**Line 376-378: this sentence is confusing, please simplify it**

We rephrased the sentence as follows:

*Sorption of proteins on clay and Fe-mineral surfaces is usually highest close to the isoelectric point of a specific protein.*

**Line 405-420: Do you mean that the climate factors influence soil pH and then regulate the depolymerization rates? However, your data doesn't seem to support this, please explain it.**

We added the following sentence:

*The correlation coefficient between MAT and depolymerization significantly decreased by removing the effects of soil Fe- and Al–oxyhydroxides, while the decrease by removing effects of soil pH was not significant.*

**Line 462: It's confusing, 'peptidase activity is a proxy of microbial N or N limitation'?**

We rewrote this sentence as follows:

*The amount of NaOH-extractable protein was here identified as the most important direct predictor of protein depolymerization rates, while peptidase activity negatively related to protein depolymerization, and therefore , rather reflects a proxy of microbial N limitation according to enzyme allocation theory (Allison et al., 2010)*

**Fig 5: 'black arrows' are missing in the figure. And how to identify the direct and indirect effects?**

We corrected the figure caption.

**2.3. Reviewer comment #2**

**180 - I don't understand what the other factor, besides land use type, is in these models. More broadly, this analysis scheme doesn't make too much sense to me... your conclusions are that bedrock type is a key driver of depoly rates, and land use type is not. But bedrock type was only subject to a 1-way anova, while land use is subject to a 2-way anova which controls for bedrock type/climate. Why, for example, was the effect of bedrock type not analyzed with a 2-way ANOVA that controlled for the effects of land use? Given relatively low sample #s, it is unsurprising that there is not enough statistical power to detect an effect of land use type in a 2-way ANOVA when controlling for bedrock/climate/soil type, but the bedrock type analysis was not subject to the same dilution of statistical power, so it seems to me that the conclusions are drawn from incommensurable statistical approaches.**

We rephrased the paragraph as follows:

*Land use effects on process rates and soil properties were analyzed for the 22 sites where cropland, grassland and woodland soils could be sampled in close vicinity (66 data points).  Since only one composite sample was analyzed per land use at each site and therefore single observations were not independent, 'site' was included as factor in a two-way ANOVA to account for differences between sites (climate, bedrock, soil type). . Given the low (non-significant) land use effects across sites the effects of bedrock were analyzed by one-way analysis of variance (ANOVA) followed by Tukey HSD tests. Not accounting for land use here allowed to analyze the whole data set (n=91) instead of restricting this to the 22 site data set (n=66).*

**205- Are the +/- numbers one standard error of the mean? Confidence interval? Please state at the first instance.**

We will indicated this at the first instance.

**360 - I have a hard time wrapping my head around how Fe and Al oxyhydroxides can simultaneously increase SOM stabilization AND increase SON availability.**

We rephrased the paragraph as follows:

*Therefore the overall organic N pool size is expected to be larger in fine textured soils and in soils high in Fe- and Al-oxyhydroxides. Moreover, the strength of the binding interaction between Fe- and Al-oxyhydroxides and SOM, and more specifically with organic N including proteins, is higher by 50% than with typical clay minerals (Newcomb et al, 2017). Consequently soils rich in Fe- and Al- oxyhydroxides contain larger pools of proteolytic substrates (organic N and proteins), but these substrates can be more strongly bound and therefore be less accessible for microbial utilization. However, column experiments with embedded goethite in acidic soils revealed that sufficiently large amounts of stabilized OM can be re-dissolved by progressing percolation of dissolved OM and the subsequent exchange with adsorbed compounds such as peptides (Leinemann et al., 2018), which thereby become available for enzymatic attack and/or microbial utilization. The bioavailability of oxide-bound organic N is further supported by the strong positive correlation between NaOH-extractable protein and amorphous Fe- and Al-oxyhydroxides (Table S3), since NaOH mainly extracts loosely bound proteins (Wattel-Koekkoek et al., 2001). Overall, Fe- and Al-oxyhydroxides remained as a significant parameter in linear models and path analyses and should therefore be considered as important predictors for the potential of a soil to retain and accumulate SOM (Moni et al, 2007; Fang et al., 2019), which promotes microbial biomass and activity (Xu et al., 2013; Hartman and Richardson, 2013). The positive effect of the potential to accumulate SOM can be attributed to the continuous exchange of adsorbed compounds and the consequent steady release of organic N. ". The net effect of these adverse interactions is currently unknown; therefore this study is among the first to show a net positive effect of Fe- and Al oxyhydroxides on the in situ rates of depolymerization of high molecular weight-ON substrates.*

**405 - "As demonstrated by partial correlations..." This statement makes an assumption that Al/Fe oxyhydroxides and pH are the TRUE controls, and MAT is just a latent predictor, which I don't think has been fully justified.**

We added the following sentence:

*The correlation coefficient between MAT and depolymerization significantly decreased by removing the effects of soil Fe- and Al–oxyhydroxides, while the decrease by removing effects of soil pH was not significant.*